

# Coccolith arrangement follows Eulerian mathematics in the coccolithophore *Emiliania huxleyi*

Kai Xu[1,2], David Hutchins[3] and Kunshan Gao[2]

[1] College of Fisheries, Jimei University, Xiamen, Fujian, China
[2] State Key Laboratory of Marine Environmental Science, Xiamen University, Xiamen, Fujian, China
[3] Department of Biological Sciences, University of Southern California, Los Angeles, CA, USA

## ABSTRACT

**Background:** The globally abundant coccolithophore, *Emiliania huxleyi*, plays an important ecological role in oceanic carbon biogeochemistry by forming a cellular covering of plate-like $CaCO_3$ crystals (coccoliths) and fixing $CO_2$. It is unknown how the cells arrange different-sized coccoliths to maintain full coverage, as the cell surface area of the cell changes during daily cycle.

**Methods:** We used Euler's polyhedron formula and CaGe simulation software, validated with the geometries of coccoliths, to analyze and simulate the coccolith topology of the coccosphere and to explore the arrangement mechanisms.

**Results:** There were only small variations in the geometries of coccoliths, even when the cells were cultured under variable light conditions. Because of geometric limits, small coccoliths tended to interlock with fewer and larger coccoliths, and vice versa. Consequently, to sustain a full coverage on the surface of cell, each coccolith was arranged to interlock with four to six others, which in turn led to each coccosphere contains at least six coccoliths.

**Conclusion:** The number of coccoliths per coccosphere must keep pace with changes on the cell surface area as a result of photosynthesis, respiration and cell division. This study is an example of natural selection following Euler's polyhedral formula, in response to the challenge of maintaining a $CaCO_3$ covering on coccolithophore cells as cell size changes.

Corresponding author
Kunshan Gao, ksgao@xmu.edu.cn

## INTRODUCTION

The structure of a normal, complete *Emiliania huxleyi* coccolith includes two oval-shaped shields connected by a central tube (*Young et al., 1992*). The coccoliths are curved and interlocked with neighboring coccoliths to match the spherical morphology of the cell membrane (*Young, Bown & Lees, 2017*; *Young et al., 2003*). Generally, in healthy, rapidly growing *E. huxleyi* cells, a layer of interlocking coccoliths fully and smoothly covers the protoplast surface, forming the so-called coccosphere. Thus, the coccosphere is generated when a cell arranges a group of interlocking coccoliths to fully cover the cell

surface. In addition, the *E. huxleyi* coccoliths vary in size among morphotypes, strains, within strain-specific populations, and even frequently observed on individual cells (*Paasche, 2001*). Although the interlocking coccolith architecture can offer exceptional mechanical protection for *E. huxleyi* cells (*Jaya et al., 2016*), the coccolith topology (the relationship between interlocking coccoliths of the coccosphere) and the arrangement mechanisms remain unknown.

Numerous studies have reported that the cell topology of many organisms follows mathematical rules. The two-dimensional (2D) Euler's formula was used in previous studies to explain why the average number of cell sides is six in many tissues, such as plant coverings, animal epithelia, and seaweed (*Gibson et al., 2006*; *Lewis, 1926*; *Xu et al., 2017*). The three-dimensional (3D) Euler's formula was used to explain why the average face number of cells is nearly 14 in soap froth and many multicelled organisms (*Lewis, 1943*; *Weaire & Rivier, 1984*). As the basic component of the coccosphere, coccoliths are produced with a specific geometry. Thus, we propose that the formation of the coccosphere must follow some basic mathematical principles or constraints. Understanding the mathematical controls and limits of coccolith topology would be extremely useful for modeling the architecture of some extinct coccolithophore species that were rarely preserved in the fossil record, which were observed only as loose coccoliths and never as intact coccospheres (*Sheward, 2016*). In addition, to sustain full coverage of the cell surface by coccoliths as cell cycle induces changes, there must be a link between the coccolith number and cell size. In the present study, we used Euler's polyhedron formula and CaGe simulation software, validated by the geometries of coccoliths and the coccosphere, to investigate the mathematical constraints that might underpin the coccolith topology in the *E. huxleyi* coccosphere.

## MATERIALS AND METHODS

The coccosphere diameters and geometric data of coccoliths which are presented in this study were derived from a previous study by *Xu & Gao (2015)*. Briefly, *E. huxleyi* calcifying strain CS–369 was grow in Aquil medium (*Price et al., 1989*) at 20 °C at two $CO_2$ concentrations (400 and 1,000 ppmv). The cultures were exposed to either artificial light (12 h:12 h light:dark), or solar irradiance (14 h:10 h light:dark) with and without an ultraviolet screen. The mean visible light levels during the light period were ranged from ~100 to 650 $\mu$mol m$^{-2}$ s$^{-1}$. Cultures collected at the same time point during the light period were filtered gently onto 1 $\mu$m polycarbonate (*Xu & Gao, 2015*) or 0.22 $\mu$m mixed cellulose ester filters. We examined and imaged these filters with a Philips XL30 (Philips, Eindhoven, the Netherlands) (*Xu & Gao, 2015*) (Amsterdam, The Netherlands) or an LEO 1530 (Carl Zeiss SMT GmbH, Oberkochen, Germany) scanning electron microscope (SEM). The morphology of *E. huxleyi* coccoliths was characterized as follows: normal, incomplete, malformed, incomplete, and malformed (*Langer et al., 2011*; *Xu & Gao, 2015*). We selected detached normal coccoliths lying flat on the filters to measure the distal shield length (DSL), distal shield width (DSW), and outer distal shield width (OSW) using the software Amscope Toupview 3.0 (Fig. 1). All of the coccolith samples from the different growth conditions were combined to explore the general mathematical

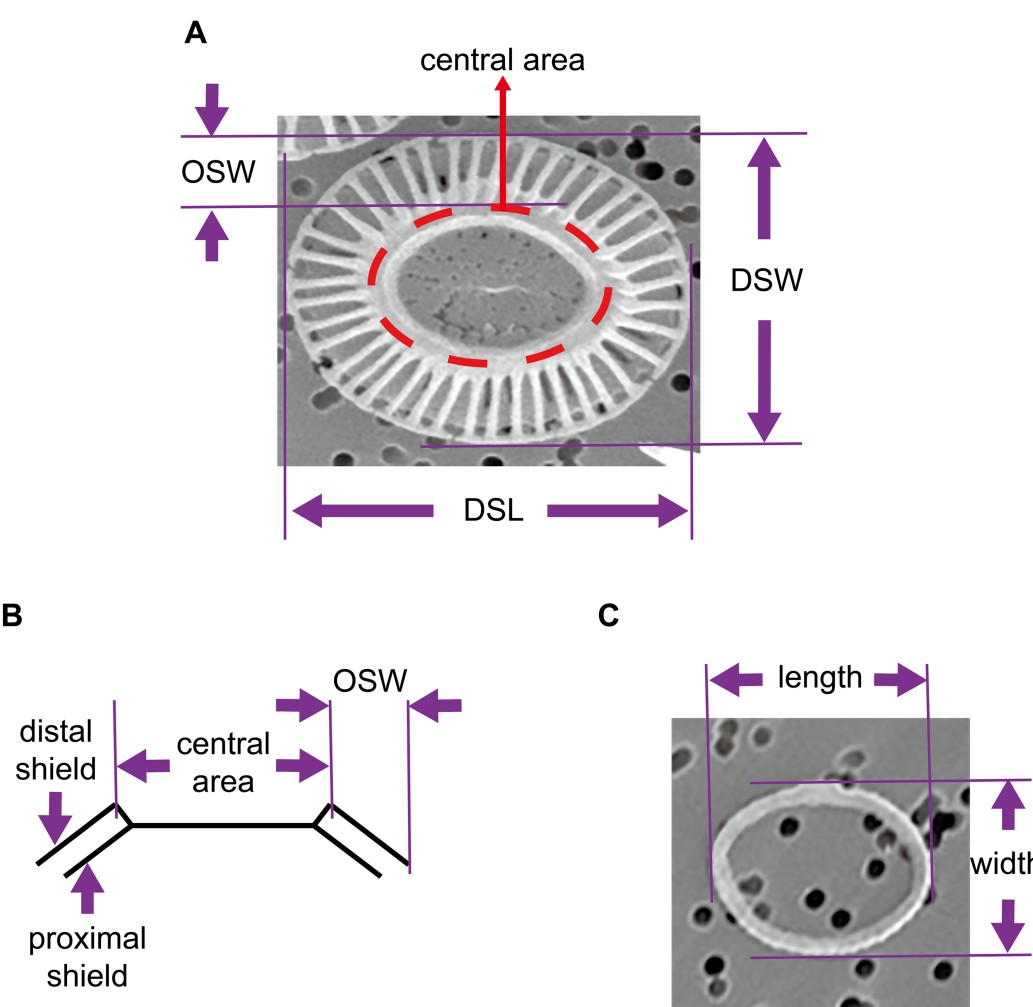

**Figure 1 Geometry of coccoliths and proto-coccoliths.** (A) A frontal view of a coccolith. (B) A diagram of a side view of a coccolith. Angle α refers to the angle between the distal shield and central area. DSL, DSW and OSW in front view (A) and side view (B) of coccolith. Length and width of proto-coccolith (C).

principles underlying mechanisms of coccolith arrangement on the surface of the protoplast. In addition, this study also measured the length and width of oval-shaped proto-coccoliths lying flat on filters. The proto-coccolith ring is the initial calcite crystal of coccolith and was previously identified as an incomplete coccolith (*Langer et al., 2011*; *Paasche, 2001*; *Xu & Gao, 2015*; *Young et al., 1992*).

We define a layer of coccoliths as a group of completely interlocking coccoliths that fully covered the cell surface. In the present study, except for a few extra non-interlocked coccoliths, most visible coccoliths were completely interlocked with each other. Thus, the coccospheres examined in this study contained only one layer of coccoliths. Assuming that the coccoliths were uniformly distributed, the number of coccoliths in the coccosphere was equal to twice that of the fully interlocked coccoliths with a visible central area on the SEM photos (Fig. 1). In addition, we measured the coccosphere diameter to establish a relationship between coccolith number per coccosphere and surface area of

coccosphere. We analyzed the structures of the coccospheres using Euler's polyhedral formula, where a polyhedron with $F$ faces, $V$ vertices and $E$ edges is described by $F + V - E = 2$. Then, we examined and simulated data using CaGe (https://www.math.uni-bielefeld.de/CaGe/) (*Brinkmann et al., 2010*). We used SPSS 22 to test the normality of data and we applied R 3.4.3 (http://cran.r-project.org) to calculate the probability of coccoliths with specific geometric characteristics.

## RESULTS AND DISCUSSION

### Coccospheres contain only one layer of coccoliths

Extra coccoliths attached on inner coccoliths have frequently been found on the coccospheres of *E. huxleyi*, which led to the idea that *E. huxleyi* cells contain "multilayers" of coccoliths (*Hoffmann et al., 2015*; *Paasche, 2001*; *Sviben et al., 2016*; *Young et al., 2003*). A previous study found that the movement of the protoplast membrane of the coccolithophore *Coccolithus pelagicus* caused the coccoliths to interlock with each other (*Taylor et al., 2007*). Thus, extra coccoliths were beyond the control of coccolithophore cells because they were not directly attached to the protoplast membrane. These extra coccoliths could not completely interlock with each other and were not able to fully cover the surface, allowing them to easily detach from the cells. In general, many detached coccoliths are found in the medium when culturing *E. huxleyi* (*Paasche, 2001*). In this study, a layer of coccoliths was defined as a group of completely interlocking coccoliths that fully covered the cell surface. Therefore, the *E. huxleyi* cells should contain only one layer of completely interlocked coccoliths, which enables full coverage by coccoliths on the protoplast surface.

### Number of bordering coccoliths per coccolith

The geometries of coccoliths from *E. huxleyi* CS-369 were microscopically measured (Fig. 1). The ranges of the coccosphere diameter, DSL, and DSW were found to be 3.92–8.04 μm (5.54 ± 0.63 μm), 2.05–4.38 μm (3.04 ± 0.40 μm), and 1.34–3.92 μm (2.47 ± 0.38 μm), respectively (Table 1). The average ratio of DSW/DSL was 0.81 ± 0.07 ($n = 1,918$). The shields were ellipsoid-ring-shaped structures, and the space between two shields enabled the coccoliths to overlap at the shield area (Figs. 1 and 2). The coccolith layer of *E. huxleyi* exhibited two distinguishing features: every two neighboring coccoliths (faces) were overlapped in shields, with every three overlaps (edges) intercrossed at a junction (vertex). The edges formed an inscribed polygon, placed inside the coccolith, with each vertex on the circumference (the big ellipse) of the coccolith (Fig. 3).

The OSW ranged from 0.36 to 0.84 μm (0.57 ± 0.09 μm), and the ratio of OSW/DSL ranged from 0.15 to 0.25 (0.19 ± 0.02) (Table 1). The calculated length and width of the central areas were, respectively, 1.90 and 1.31 μm, which matched the measured length (1.86 μm) and width (1.37 μm) of the proto-coccolith rings very well. These results were consistent with a previous study, which found that the position of the proto-coccolith ring corresponded to the base of the central tube (*Young et al., 1992*). The distal and proximal shields are connected by the central tube (*Young et al., 1992*), which indicates that the overlap of interlocking two coccoliths could not transects through the

**Table 1 Geometric data for coccolith and proto-coccolith, coccosphere diameter, and coccoliths per cell.**

| Sample | | Mean ± SD | Range | $n$ |
|---|---|---|---|---|
| Normal coccoliths | Distal shield length (DSL, μm) | 3.04 ± 0.40 | 2.05–4.38 | 1,918 |
| | Distal shield width (DSW, μm) | 2.45 ± 0.37 | 1.34–3.92 | 1,918 |
| | DSW/DSL | 0.81 ± 0.07 | 0.62–0.99 | 1,918 |
| | Outer distal shield width (OSW, μm) | 0.57 ± 0.09 | 0.36–0.84 | 70 |
| | OSW/DSL | 0.19 ± 0.02 | 0.15–0.25 | 70 |
| Proto-coccoliths | Length (μm) | 1.86 ± 0.21 | 1.45–2.38 | 72 |
| | Width (μm) | 1.37 ± 0.20 | 0.98–1.90 | 72 |
| | Length/width | 0.74 ± 0.09 | 0.56–0.94 | 72 |
| Coccospheres | Coccosphere diameter (μm) | 5.54 ± 0.63 | 3.92–8.04 | 156 |
| | Coccolith number per cell | 15.4 ± 3.6 | 6–30 | 156 |

central area. According to observations in previous studies and this study, normal coccoliths are always interlocked very closely (*Paasche, 2001*; *Young et al., 2003*). Thus, the coccoliths have the highest possible overlap, which could help their constant attachment to the cell surface.

By simulating the interlocking patterns on 2D planes, this study summarized three basic principles:

1. The edges of the inscribed polygon must not transects through or intersects with the central area (Figs. 1 and 2). Because of the restriction of central tube, it is impossible for a coccolith to have an inscribed triangle (Table 1; Fig. 3). The ratios of OSW/DSL were normally distributed. Only when the OSW/DSL of a coccolith increased to $\geq 0.37$ (probability $<10^{-15}$) did sufficient space in the shield area enable that coccolith to interlock with three much bigger coccoliths (Fig. 4).

2. Small coccoliths tended to interlock with fewer and larger coccoliths, and large coccoliths tended to interlock with more and smaller coccoliths (Fig. 4). The second principle is very likely to be a combination of Lewis's law and the Aboav–Weaire Law. According to Lewis's law, if polygons are tessellated on a 2D plane, then small polygons tend to have fewer sides (linear relationship between mean area of a $n$-sided polygon with $n$) (*Lewis, 1928*; *Weaire & Rivier, 1984*). According to Aboav–Weaire Law, polygons with large numbers of sides tend to have few-sided neighbors (*Weaire & Rivier, 1984*). However, this principle does not mean that a greater size difference between the central coccolith and the surrounding coccoliths is better for coccolith interlock. A large ratio between the average size of the bordering coccoliths ($DSL_{BCs}$) and the size of the central coccolith ($DSL_{CC}$) resulted in overlaps that transect through central areas, and small $DSL_{BCs}/DSL_{CC}$ resulted in loose interlock (Fig. 5).

3. The coccoliths with fewer edges tended to have high ratios of DSW/DSL (Fig. 3) and OSW/DSL (Figs. 3 and 4), but the coccoliths with more edges were less sensitive to these two ratios (Figs. 3–5). If coccoliths had fewer edges, then the edges tended to be close to the center of coccolith, and consequently, high ratios of DSW/DSL, and

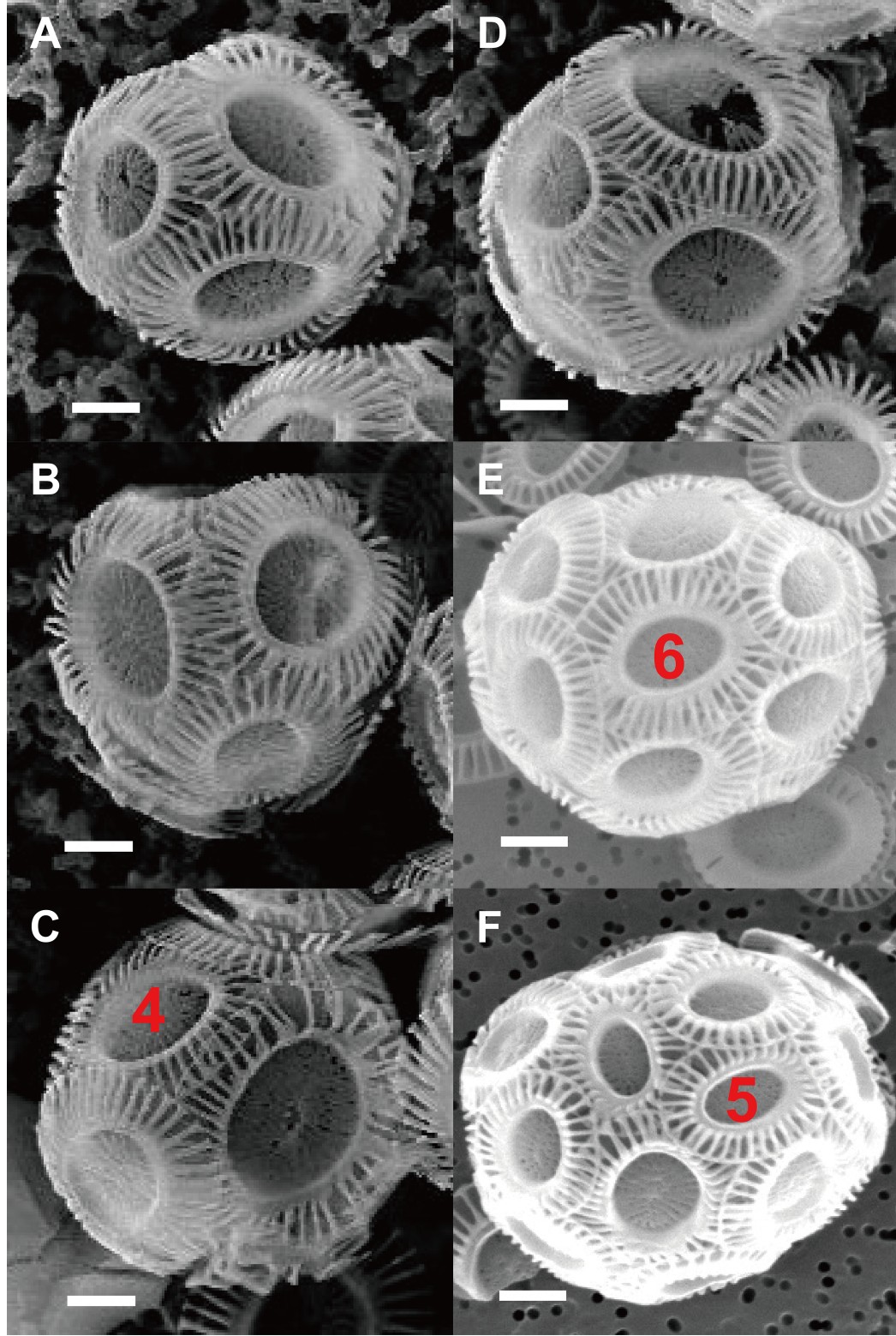

**Figure 2 Photos of coccospheres of *E. huxleyi.*** Coccospheres with different numbers of coccolith: 6 (A), 8 (B–C), 10 (D), 14 (E), 22 (F) coccoliths. White bars are 1 μm. Red numbers on the coccoliths indicate bordering coccoliths.

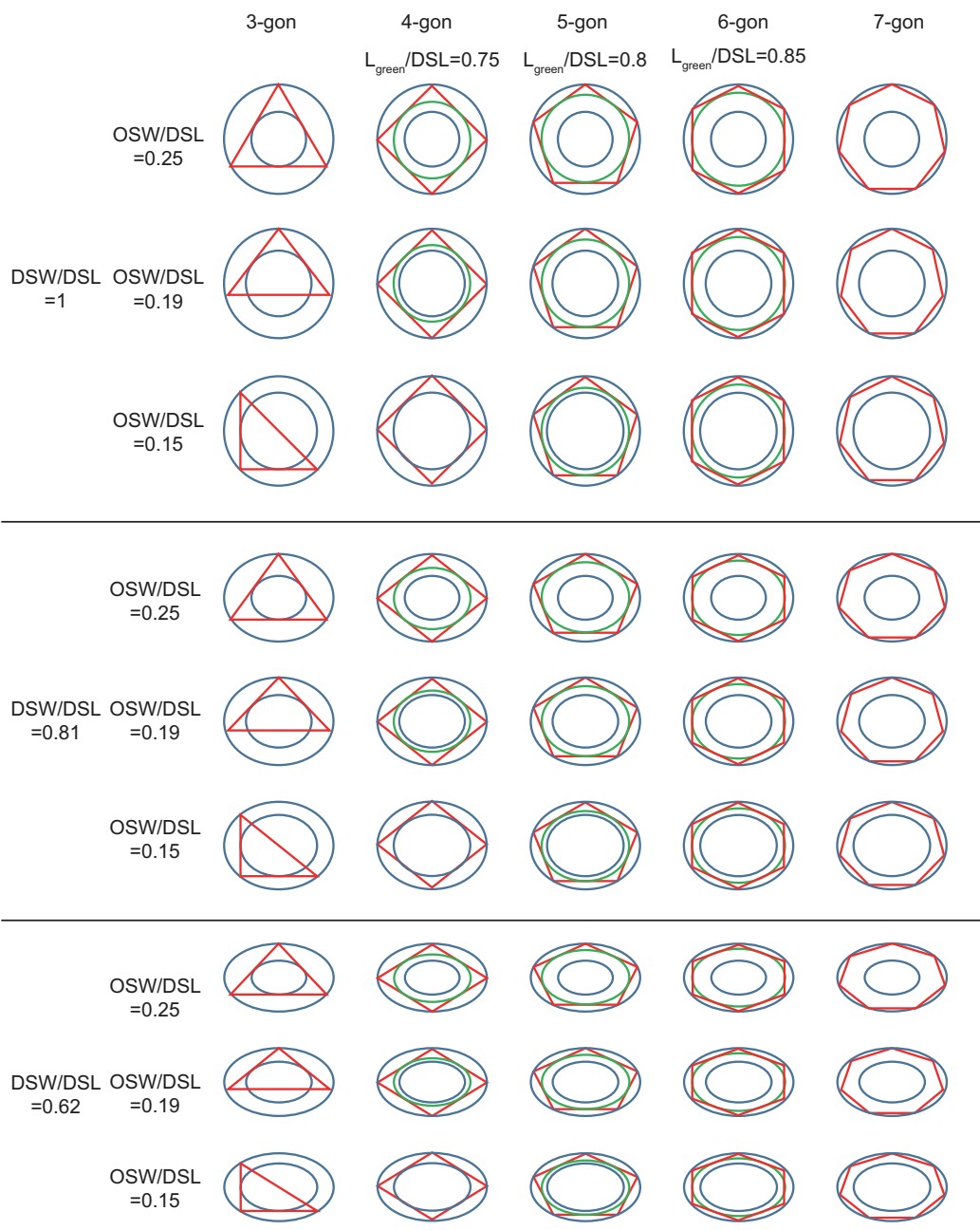

**Figure 3 Inscribed tri- to heptagons in coccoliths.** Large and small blue ellipses of oval-shaped rings represent edges of distal shield and central area of coccoliths, respectively. Maximum, average, and minimum values of DSW/DSL and OSW/DSL are derived from Table 1. Green ellipse represents the maximum inscribed ellipse of the maximum inscribed polygon (in red color) of coccolith. The ratio of length of green ellipse to DSL appear above the photos.

OSW/DSL are needed to provide sufficient space for overlaps (Figs. 3 and 4). For coccolith with more edges, however, the edges were far away from the center, and then the overlaps were less sensitive to the size of the central area and ratio of DSW/DSL (Figs. 3–5). In addition, coccolith with smaller OSW can be fully overlapped in the shield area, and coccolith with larger OSW would be only partly overlapped (Figs. 3–5).

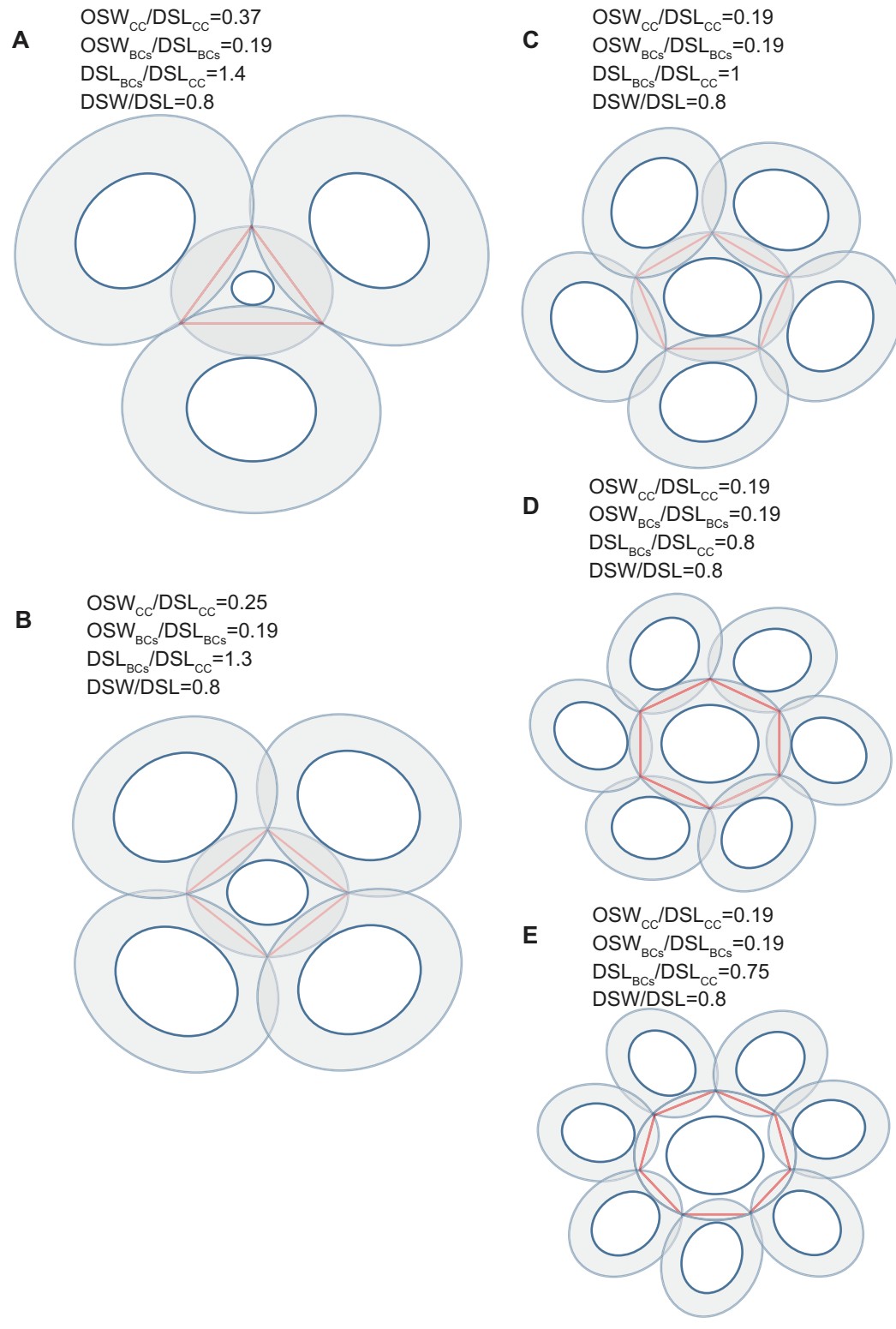

**Figure 4 Simulation of interlocked coccoliths on a 2D plane.** Central coccolith interlocked with three, four, five, six, and seven others (A–E). DSW/DSL of all coccoliths were set to an average value of 0.81. Values of $OSW_{CC}/DSL_{CC}$, $OSW_{BCs}/DSL_{BCs}$, and $DSL_{BCs}/DSL_{CC}$ appear above the photos.

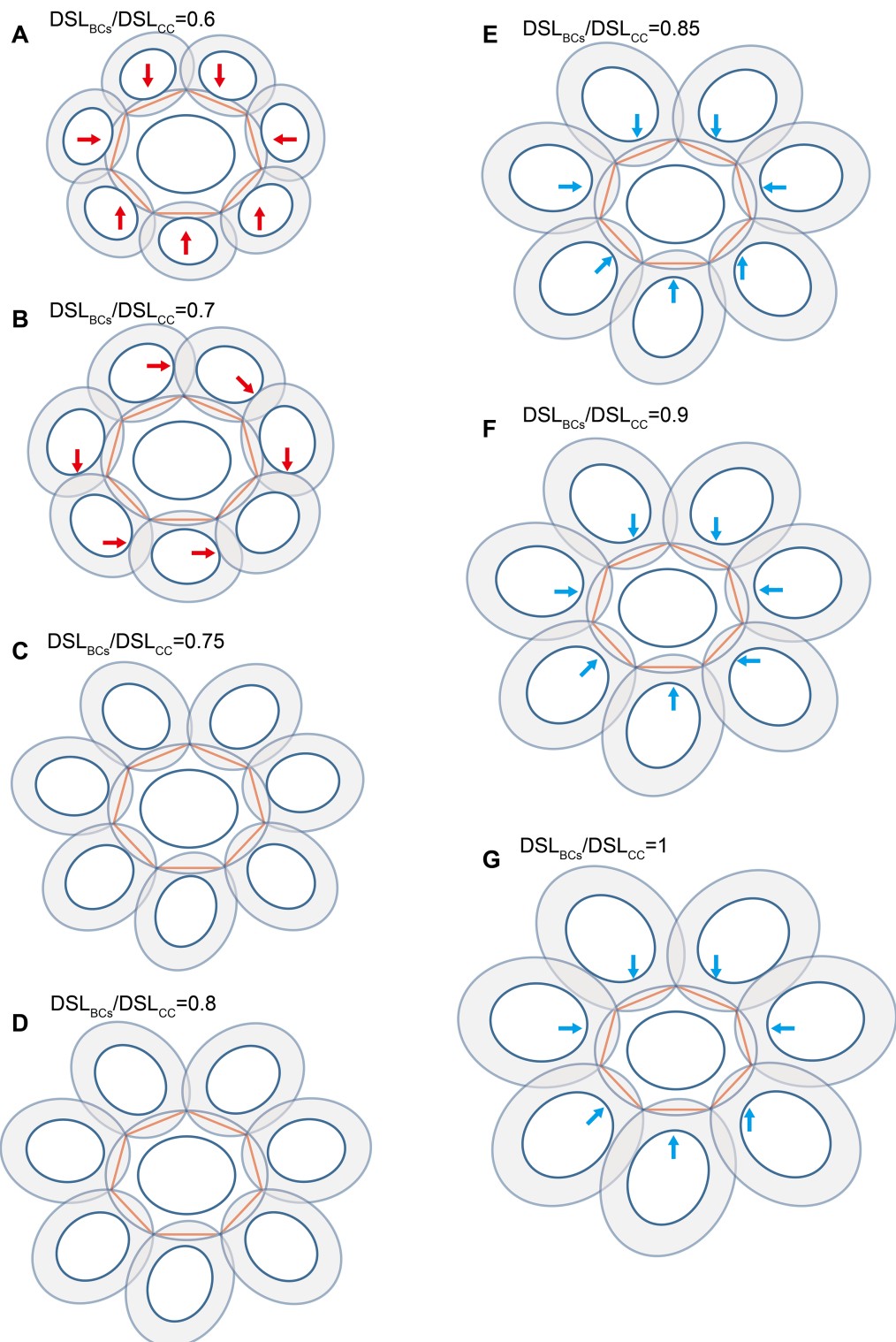

**Figure 5 Each coccolith interlocked with seven others.** DSW/DSL and OSW/DSL were set at average values of 0.81 and 0.19, respectively. Values of $DSL_{BCs}/DSL_{CC}$ appear above photos. Red arrows indicate that overlaps transect the central area of the central coccolith (A), or the bordering coccoliths because of they should not on the same plane of the central coccolith (B). The value range of $DSL_{BCs}/DSL_{CC}$ of the ideal interlocking patterns are 0.75–0.8 (C–D). Blue arrows indicate loose interlock of bordering coccoliths (E–G).
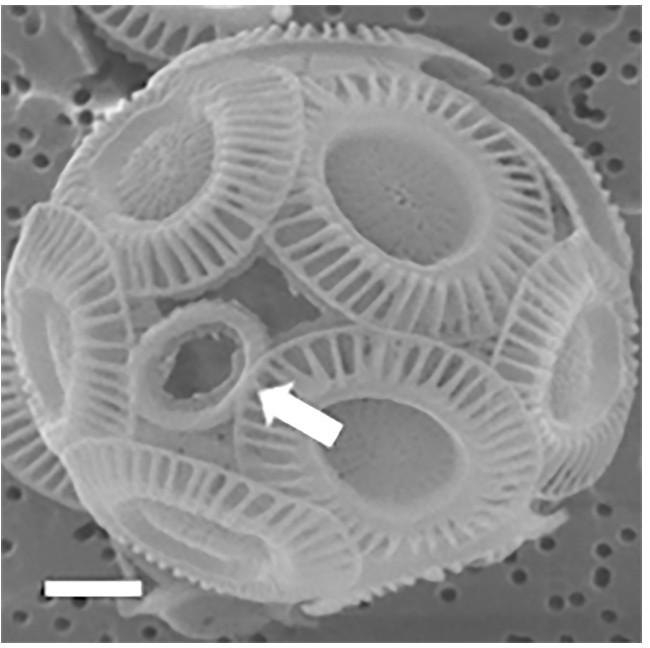

**Figure 6 Coccosphere of *E. huxleyi* with an incomplete coccolith.** White bar is 1 μm and white arrow demonstrates incomplete coccolith.

Our simulation found that polygons with four or more edges could be inscribed in the shield area of coccoliths and meet the noted three principles (Fig. 3). Thus, each coccolith must have four or more interlocking coccoliths. The distributions of the DSL and DSW were normal, and 90% of the data were clustered in the ranges of 2.41–3.74 μm, and 1.90–3.11 μm, respectively. The 5% and 95% percentiles of DSL were 21% smaller and 23% larger than the mean, respectively. Thus, for the strain used in this study, the coccolith sizes had only small variations even when grown under different conditions. Assuming that a coccolith interlocks with seven others, then $DSL_{BCs}$ would need to be approximately 0.75–0.8 times $DSL_{CC}$ (Fig. 5). The probability of such a group of eight coccolith with $0.75 \leq DSL_{BCs}/DSL_{CC} \leq 0.8$ is $4.4 \times 10^{-10}$. On the basis of these analyses, the greatest possible number of edges per coccolith is six.

The observation of SEM photos also supported these conclusion: each coccolith of cells of the coccolithophore *E. huxleyi* interlocks with four to six bordering coccoliths (Fig. 2). *E. huxleyi* can sometimes produce and incorporate incomplete or malformed coccoliths into coccospheres under field and laboratory conditions (*Paasche, 2001*; *Xu & Gao, 2015*). The non-normal coccoliths can cause gaps between coccoliths (*Xu & Gao, 2015*) (Fig. 6). In addition, incorporating of too many abnormal coccoliths may inhibit the formation of the coccosphere and compromise its defensive abilities (*Langer & Bode, 2011*).

## Analyze coccolith topology using Euler's formula and the CaGe software

The number of coccoliths in spherical coccospheres of *E. huxleyi* varies widely, ranging from six to 30 coccoliths per cell (Fig. 2; Table 1). The 3D structures of coccospheres are

typically cubic polyhedrons. The convex polyhedron meets Euler's polyhedral formula $F + V − E = 2$, which can be expressed as:

$$\sum (6 − n)F_n = 12 \qquad (1)$$

where $F_n$ means the number of faces with $n$ edges ($4 \leq n \leq 6$). This equation can be simplified as $2F_4 + F_5 = 12$, where $F_4$ and $F_5$ are the number of four- and five-edged (interlocking with four and five coccoliths) coccoliths, respectively (*Grünbaum & Motzkin, 1963*). Thus, the lowest coccolith number necessary to construct a complete coccosphere is six (Fig. 2; Table 1). In addition, because the equation does not set any restriction on the number of hexagons, the necessary conditions of Euler's polyhedral formula are insufficient for use in the enumeration of polyhedra (*Grünbaum & Motzkin, 1963*) (Table S1). In this study, we applied the simulation software CaGe to test for the existence of polyhedra, as deduced from Eq. (1).

A polyhedron is a solid that contains at least four faces. Assuming a coccosphere only contains four or five coccoliths (Fig. 7), then some coccoliths should have only three edges, which would result in the average angles between neighboring coccoliths being less than 90°. In this way, the coccoliths must be far longer than the diameter of the protoplasm, which is not consistent with the fact that the calcification of *E. hulxeyi* at the diploid phase is an intracellular process (*Dixon, 1900*; *Paasche, 2001*; *Wilbur & Watabe, 1963*). This fact also supports the idea that the lowest coccolith number to construct a complete coccosphere is six (Fig. 2; Table 1), which in turn also results in that each coccolith must having four or more interlocking coccoliths. This conclusion is consistent with previous studies that have observed that other coccolithophore species, *Helicosphaera carteri*, *Toweius pertusus*, and *Umbilicosphaera bramletti*, have minimum numbers (five to seven) of coccoliths per cell (*Gibbs et al., 2013*; *Sheward, 2016*; *Sheward et al., 2017*; *Young, Bown & Lees, 2017*).

Moreover, according to our calculations, if the coccolith number per coccospheres is 12 or 14 or more, then the polyhedra (coccospheres) must contain five- and six-edged faces; if the number is 13 or 12 or fewer, then they must contains four- and five-edged faces (Fig. 7; Table S1). In addition, the polygons with four and five edges betray the positive overall curvature (*Weaire & Rivier, 1984*), which indicates that the polygon composition influences the curvature of spherical cell.

**Effective coverage area of coccolith**

The average surface area of a coccosphere was 97.5 $\mu m^2$. The mean DSW was very close to the mean DSL (Table 1), and then the surface area of coccolith approximately equal to the surface area of a spherical cap, with a bottom diameter equal to the mean value of DSW and DSL. The estimated surface area of a coccolith was 6.3 $\mu m^2$ which was slightly larger (6%) than the projected area of a coccolith on a 2D plane. Thus, we used the 2D geometric laws to approximately estimate the effective coverage area of coccolith (the area of the inscribed polygon).

The area of maximum inscribed and minimum circumscribed polygons of an ellipse are, respectively, $0.5nab\sin(2\pi/n)$ and $nab\tan(\pi/n)$, where $a$ and $b$ are the two semi-axes

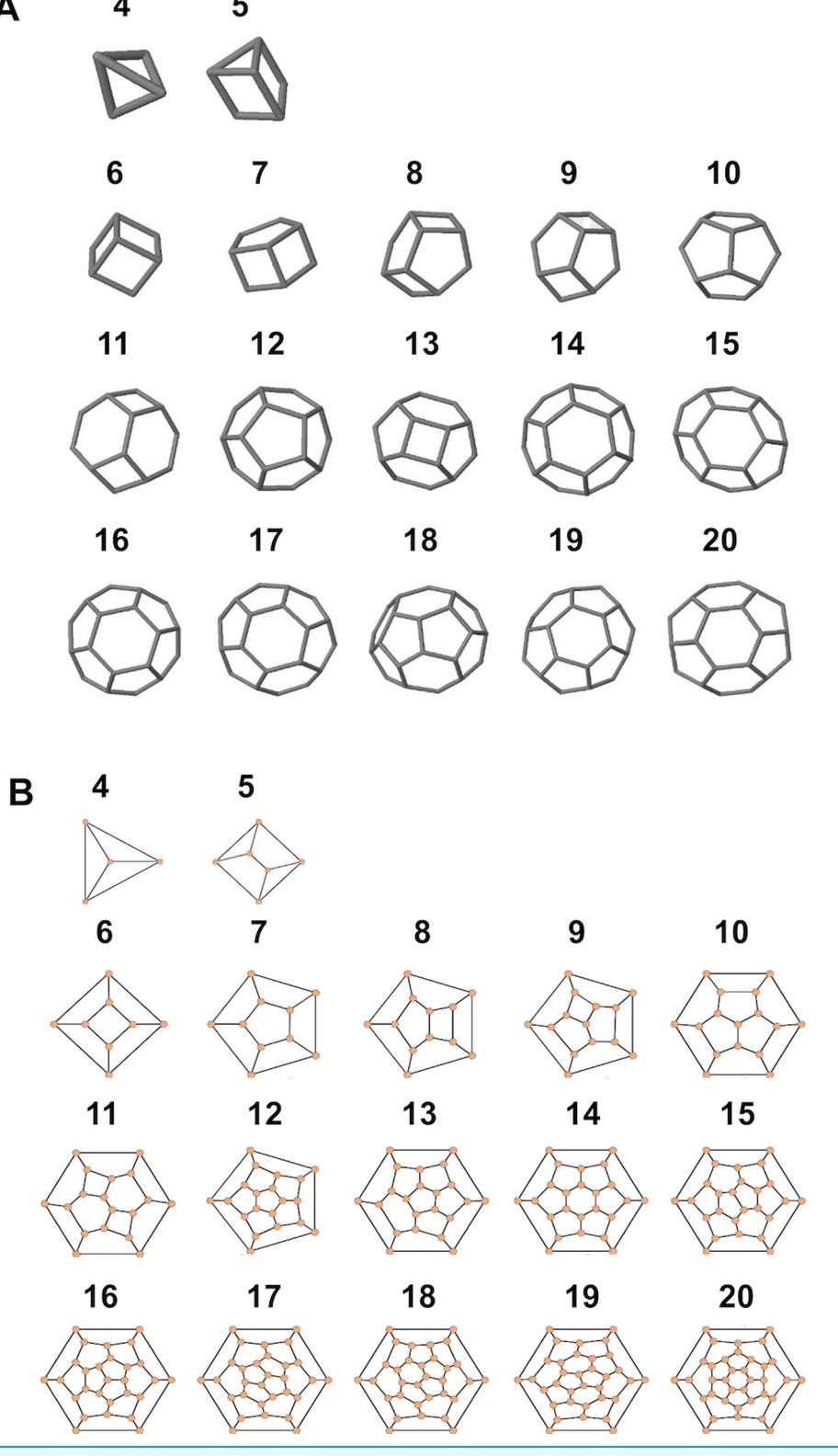

**Figure 7 Simulated structures of typical coccospheres.** 3D (A) and 2D (B) structures of polyhedra simulated by CaGe. The number of faces appears above the structures. Polygon compositions and isomer numbers are shown in Table S1.

of the ellipse, $n$ is the number of sides (*Su, 1987*). Thus, the area of maximum inscribed ellipse of the maximum $n$-sided inscribed polygon of a coccolith is $(\cos(\pi/n))^2$ times the area of coccolith. The mean number of coccoliths per coccosphere was 15.4 which results in that the mean edge number of coccolith was 5.2 (Table 1). Consequently, axes of the maximum inscribed ellipse were approximately 0.82 times the axes (DSL and DSW) of coccoliths, which equalled to the mean axes of coccolith and proto-coccolith. Thus, the key geometric constraint ensuring coccolith maximize its coverage was sustaining the ratio of OSW/DSL at 0.18. On the other hand, coccolith with large OSW/DSL requires more $CaCO_3$ because the shield area needs more $CaCO_3$ than the central area. Obseved values of OSW/DSL matched very well with calculated values (Table 1), which indicated that coccoliths were produced with minimum $CaCO_3$ under the premise of maximizing the coverage area.

The ratio of the area of the maximum inscribed polygon of ellipse to the ellipse area is:

$$0.5 \times n/\pi \times \sin(2\pi/n) \tag{2}$$

where $n$ is the number of sides of polygon. Thus, the effective coverage area of a coccolith increases with an increasing number of bordering coccoliths. We used the polygon composition of coccosphere (Table S1) and Eq. (2) to estimate the mean number of coccoliths per coccosphere. The estimated number was 19.7 which is higher than the observed number by 22%. Thus, the method in this study underestimated the number of coccoliths per coccosphere, which could attribute to that a few coccoliths with hidden central area were not counted in the SEM photos, or coccosphere size was increased by the preservation processes. The estimated effective coverage area of coccolith was 4.9 $\mu m^2$. In addition, the correction on the mean number of coccoliths per coccosphere did not influence the conclusion in the above paragraph.

## Coccoliths per coccosphere and daily size changes

Regardless of the size difference of coccospheres, the cells can always maintain full coverage by coccoliths (Fig. 2). Full coverage by coccoliths is sustained even during cell division (*Klaveness, 1972*). The lowest coccolith number per coccosphere is six, which raises a question of how *E. huxleyi* cells sustain full coverage after division. The daughter cells may lose full coverage of coccoliths if division is triggered with less than 12 coccoliths per coccosphere. The cell, however, cannot to count the number of coccoliths. This study found that larger coccospheres of *E. huxleyi* used more coccoliths to cover their larger surface area ($p < 0.0001$, Fig. 8), which was consistent with several recent studies (*Gibbs et al., 2013*; *Hoffmann et al., 2015*; *Sheward, 2016*; *Sheward et al., 2017*). Cell size is tightly regulated during the whole cell cycle to maintain the characteristic cell size of a population (*Amodeo & Skotheim, 2016*; *Kiyomitsu, 2015*). Thus, we propose that *E. hulxeyi* cells divide at the proper size, which ensures that sufficient coccoliths will be allocated to the daughter cells, that is, each daughter will obtain at least six coccoliths.

Based on observations of four other coccolithophore species (*Calcidiscus leptoporus*, *Calcidiscus quadriperforatus*, *H. carteri*, and *Coccolithus braarudii*), *Sheward et al. (2017)*

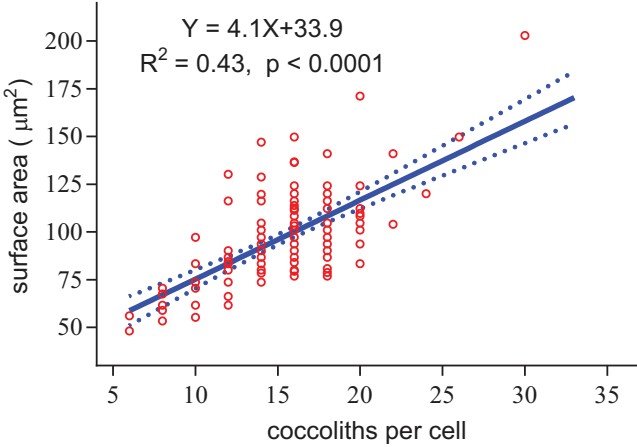

**Figure 8 Positive linear relationship between coccoliths per cell and coccosphere surface area.**

found that the cell size and the number of coccoliths per cell of recently divided cells were smaller than for cells ready to divide. Thus, to sustain a full coverage on the protoplast, calcification must adjust to changes in the surface area. During the light period, the surface area will increase as organic carbon is fixed, resulting in cells requiring more coccoliths to maintain full coverage. Calcification primarily takes place during the light period, which sequentially produces coccoliths that join the coccosphere (*Paasche, 2001*; *Taylor et al., 2007*). A large number of detached coccoliths have been found in exponentially growing cultures of *E. huxleyi*, which indicates that cells generally produce more coccoliths than needed to construct a coccosphere (*Paasche, 2001*). This may be mechanical insurance for *E. huxleyi* cells to sustain full coverage during the light period.

During the dark period, the surface area is mainly determined by cell division and respiration. The division of *E. huxleyi* cells primarily occurs at night, and the population growth rate of calcifying strains can be as high as approximately two divisions per day (*Paasche, 2001*; *Xu & Gao, 2015*; *Zondervan, Rost & Riebesell, 2002*), which means that cell numbers at the end of the dark period will be increased by as many as four times. Because of cell-to-cell heterogeneity in growth rate in the population (*Damodaran et al., 2015*), some cells need to divide three times per day. The total surface area would increase by approximately 58% if a cell equally split into four daughter cells after two sequential divisions. Previous studies reported that respiration decreased *E. huxleyi* cell size and increased the ratio of calcite to organic carbon content during the dark period (*Paasche, 2001*; *Zondervan, Rost & Riebesell, 2002*). These findings suggest that respiration could offset the effects of cell division on surface area. In addition, coccolithophores slowly calcify in the dark (*Paasche, 2001*; *Taylor et al., 2007*), which also offset these effects.

The slope of the fitted line for surface area of coccosphere and number of coccoliths per coccosphere was 4.1 (Fig. 8). The effective coverage area of coccolith was 4.9 $\mu$m$^2$. The number of coccoliths per coccosphere was underestimated by 22%. Thus, the coccoliths per coccosphere can explain 73% of the variation of the surface area of coccosphere. Large coccoliths tend to have a large coverage area, which in contrast to the assumption that, in

the population of a strain of coccolithophore, there is a positive correlation between coccolith size and coccosphere surface area. But this relationship have been observed between different species and/or strains (*Gibbs et al., 2013*; *Henderiks, 2008*; *Henderiks et al., 2012*; *Sheward, 2016*). Overall, for a cell with a layer of completely interlocking coccoliths, the effective coverage area of coccoliths must adjust to the surface area.

## Coccolith topology of a specific coccolithophore

The coccolith geometry of the coccolithophore family *Braarudosphaera* is a mystery. This family is distinguished by having a coccosphere formed of 12 five-fold symmetric pentaliths (identical pentagonal faces), which are perfectly arranged into a regular dodecahedron (*Young et al., 2003*). A polyhedron with 12 pentagons is one of the possible coccolith arrangement patterns in *E. huxleyi* cells (Table S1). The regular dodecahedron is one of the five Platonic solids, thus, pentaliths cannot interlock with others. Because of the geometric constraints, a regular dodecahedron is the only choice for coccospheres of *Braarudosphaera*. This indicates that *Braarudosphaera* spp. cells cannot change sizes, because the pentalith number must be 12, and cells could not divide if the cell surface must be totally covered by pentaliths. *Braarudosphaera* spp. cells contain visible chloroplasts, but they have not been successfully grown in culture (*Hagino et al., 2013*; *Young et al., 2003*). The coccospheres of *Braarudosphaeara* do not contain perforations, which limited interactions between the cells and surrounding seawater, and *Siesser (1993)* speculated that *Braarudosphaeara* may be related to a calcareous dinoflagellate cyst. Further study based on phylogenetic analyses suggested that *B. bigelowii* belongs to the class Prymnesiophyceae, and it is most likely related to the orders Isochrysidales and Coccolithales, and two unidentified haptophytes (*Takano et al., 2006*). Therefore, we propose that the observed coccospheres of *Braarudosphaera* spp. actually belong to a resting or cyst stage of the life cycle.

## Assembliy processes of coccosphere

Although this study suggested that the coccolith topology is actually a direct mathematical solution to maintain full coverage by coccoliths on the cell surface, knowledge of how coccoliths are arranged to form a coccosphere is limited. *Taylor et al. (2007)* recorded the formation and secretion processes of three coccoliths in a completely pre-decalcified *C. pelagicus* cell. They also released a video that clearly demonstrated that the first and second coccoliths were separated by a wide distance and that the third coccolith was secreted between the first and the second. The three coccoliths were moved to arrange them in a curved-linear order on the cell surface. This suggests that the cell sensed the positions of coccoliths which directly attached on the cell membrane. Thus, the assembling process of coccosphere is under precise control, but the mechanisms remain unclear. *E. huxleyi* may use the same mechanisms as *C. pelagicus* to construct a coccosphere, because of similar coccolith morphology and interlocking patterns.

## CONCLUSION

This study determined that, because of geometric limits, small coccoliths tend to interlock with fewer and larger coccoliths to sustain a full coverage on the spherical cell surface, and vice versa. *E. huxleyi* cells arrange individual coccoliths to interlock with four to six others, resulting in each coccosphere containing at least six coccoliths. This study used Euler's polyhedron formula and CaGe simulation software, validated with the geometries of coccoliths, to demonstrate that the proposed coccolith arrangement pattern as the only mathematical solution to form coccospheres. In addition, the number of coccoliths per coccosphere must adapt to changing cell surface area due to photosynthesis, respiration, and cell division. *E. huxleyi* cells divide at the proper size to ensure that each daughter cell can obtain at least six coccoliths. Our methods may useful to analyze the coccolith topology of other coccolithophore species and cell topology of multicelled organisms. Future work is needed to determine the actual cellular mechanisms for sensing and regulating coccosphere geometry.

## ACKNOWLEDGEMENTS

We would like to thank the anonymous reviewers for the valuable improvements that they have helped us make to this manuscript. We thank LetPub (http://www.letpub.com) for its linguistic assistance during the preparation of this manuscript.

### Funding

This study was financially supported by the National Natural Science Foundation (41430967), the National Key Research Programs (2016YFA0601400), the Joint project of National Natural Science Foundation of China and Shandong Province (No. U1606404), and the Natural Science Foundation of Fujian Province (grant numbers 2016J01165). David Hutchins's visit to Xiamen University was supported by the Visiting Scholar Program of State Key Laboratory of Marine Environmental Science (Xiamen University). The funders had no role in study design, data collection and analysis, decision to publish, or preparation of the manuscript.

### Grant Disclosures

The following grant information was disclosed by the authors:
National Natural Science Foundation: 41430967.
National Key Research Programs: 2016YFA0601400.
National Natural Science Foundation of China and Shandong Province: U1606404.
Natural Science Foundation of Fujian Province: 2016J01165.
Visiting Scholar Program of State Key Laboratory of Marine Environmental Science (Xiamen University).

### Competing Interests

The authors declare that they have no competing interests.

## Author Contributions

- Kai Xu conceived and designed the experiments, performed the experiments, analyzed the data, contributed reagents/materials/analysis tools, prepared figures and/or tables, authored or reviewed drafts of the paper, approved the final draft.
- David Hutchins authored or reviewed drafts of the paper, approved the final draft.
- Kunshan Gao authored or reviewed drafts of the paper, approved the final draft.

## Data Availability

The raw data are provided in Supplemental Dataset Files.

## Supplemental Information

Supplemental information for this article can be found online at http://dx.doi.org/10.7717/peerj.4608#supplemental-information.

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
