# Peer review of "Coccolith arrangement follows Eulerian mathematics in the coccolithophore Emiliania huxleyi"

_PeerJ, doi:10.7717/peerj.4608_

## Round 0.1 · original submission · Major Revisions

The reviewers suggest that this might become an interesting paper if a lot more work and writing is performed. Please look through all the comments in their detailed reviews and revise the paper accordingly for re-evaluation.

Reviewer 1 ·

Basic reporting

The English is clear and unambiguous throughout. I've noted a few minor grammar issues under minor edits below.

The literature references are largely sufficient, although I'm not an expert in coccolith and coccosphere morphology. I have heard the idea advacned in lines 203-204 before, so I would encourage the authors to seek a reference for this idea.

The raw data are shared as part of this manuscript. The data presentation could be improved, for example by ensuring all columns have clear headings and providing a text-document summarizing the content of the data file.

The paper is self-contained and relevant to the hypotheses.

Experimental design

Within the scope of the journal.

Research question defined and meaningful. Methods well described. Some improved clarity in the research questions and flow of reasoning would improve the manuscript. See comments to the authors below.

Validity of the findings

Data robust and controlled.

Conclusions are mostly well stated -- the real conclusion is that the data support the hypothesis of interlocking liths forming polyhedra. Analysis of polyhedra topology using Euler's formula is consistent with this hypothesis. Some improvement of the exposition is required.

Additional comments

I'm not an expert in cocclithophorid biology, lith formation, or evolutionary history. My primary interest in the manuscript is from the perspective of mathematical modeling and data analysis.

The authors use a database, previously collected and described, of coccolith and coccosphere morphology in E. huxleyi, to develop a conjecture regarding the geometry of interlocking liths on the surface of the cell. Geometric constraints (encapsulated in Euler's formula for polyhedra) combined with the requirement that liths overlap in such a way to create polyhedra, imposes some constraints on how the liths are arranged on the cell surface. For example, if a particularly small cell divides with fewer than 12 liths, there will not be enough liths to form a complete coccosphere for both daughter cells. The analysis is clear, and the result is a interesting curiousity, but I'm not convinced this will provide much insight into coccolithophorid biology and coccosphere formuation.

Line 59, 60, 71. These various claims about the liths comprising a coccosphere are a bit confusing. I think you are saying that athough it is common to have "extra" liths in a second layer, you are analyzing liths in the primary coccosphere, which consists of one layer with all liths interlocking. I'm not sure if this is an assumption, but it sounds like it is supported by your data. Some reorganization of these ideas would be welcome.

Line 61. "two times the number of coccoliths with visible central area" -- given the curvature of the coccosphere, this seems like it would be an underestimate of the number of coccoliths. Liths on the outer circumference of the projected image seen by an SEM will be hard to count. For example, in the SEM labeled "6" in Fig 2. (right column, second from the top) there are two liths, one each on the left and right side of the image, that are clearly part of the coccosphere, but neither has a visible central area. Which is counted? Neither according to your description, but probably one should be counted if you are going to double the visible number to get a total number of liths per sphere. I don't know how many liths are in that coccoshpere, but I don't believe it is 14 (2 x 7). Possibly 16, or maybe a bit more. There are some partially visible ones at the bottom of the projected image as well.

line 80-81. I don't think these sentences belong with the rest of the paragraph.

I would suggest you restructure the material describing the interlocking patterns of coccoliths (lines 89-169 or so). The flow of ideas is not clear. The general argument appears to be, or perhaps should be, the following: (1) liths appear to be interlocked in the best-formed coccospheres, (2) these interlocking liths form polyhedra, (3) we can analyze the geometry of these polyhedra using Euler's formula and the CaGe software, (4) these observations yield several constraints on coccosphere morphology -- there can't be fewer than 6 liths per sphere, there can't be more than 6 liths interlocking around a central lith, a diving cell should have at least 12 liths, the topology of the sphere must (?) change when the cell divides if both daughter cells are to have well-formed coccosphere, etc.

line 91-92. I'm a bit puzzled that the average coccolith area is 5.9 µm2, coccosphere area is 120.76 µm2, which suggests about 20.5 non-overlapping coccoliths per sphere & thus presumably more liths per sphere in reality, but the average number of liths per sphere is in fact 15. Is there some postive correlation between lith size and sphere size? Or some other explanation?

line 98-100. Not well described.

line 104 and surrounding. This point is not well described.

What is the distribution of the number of interlocking coccoliths? (e.g. relative numbers of 4, 5, 6). Is this consistent with the distribution of sizes and the geometrical calculations you've done? This would be an important test to convice me that the data all hang together and are consistent with your claims.

line 112. "assumption" - this was not clearly stated earlier. The flow of ideas needs work.

line 115-117: gaps ... indicates .. efficient and fault-tolerant. Not clear what you are saying -- fault-tolerant means what in this case (what would you expect to go wrong if the coccosphere was malformed?). What does efficient mean in this case?

line 126. A proof or citation for this reformulation of the standard form of Euler's polyhedral formula would be appreciated.

lines 144-151. What is the coccolith number of a coccosphere? how does this differ from the coccolith numbers which are consecutive positive integers? I didn't understand any of this paragraph.

line 152-3. Is division of small coccospheres an observed problem?
line 153. It's unlikely that cells count liths! Other mechanisms could surely be cell size / quota.
line 152-158. This paragraph does not hang together well. Your main point is not clear and the material is not well connected.

line 162-169. This needs to be rewritten. It's not clear what you are saying.

line 185-186. "the reason why cell size... limiting light intensities" -- this is quite a different claim from the first half of the sentence. It's not clear what your argument is. Expand?

line 203-4. I've heard this claim made by others -- but I don't have a reference -- and the reasoning was different (not based on your geometric ideas). It would be worth looking to see if anyone else has published this suggestion, even with different reasoning.

Conclusions

Delete lines 230-236. Not conclusions from your work. Perhaps integrate some of this material into the introduction.

Minor edits

Abstract. Methods.
Replace "analyses" with "analyse"
Replace "arrange" with "arrangement"

Introduction.
Line 30. Split into two sentences following the citation to Young.

Materials & Methods
line 46. "Aquil" medium?

Results and Discussion
line 69. Delete "of"
line 77. Try "which will result in the outer layers not fully covering the surface"
line 140. Delete "of"
line 154. "more than" -- what? incomplete sentence.
line 156. "mechanisms" - for what process? incomplete sentence
line 160-1. This is not a complete sentence.
line 176. "full" not "fully"
line 216. Swap "mechanisms" and "behind"

Fig 2 caption
6-22: 22?

Fig 3 caption.
The claims in (a) and (e) are not at all clear. Perhaps remove these to the revised main text.

Fig 6.
Not too surprising.
What about coccolith area and coccosphere area? Do larger spheres have larger liths? Or is lith size independent of sphere size?

Reviewer 2 ·

Basic reporting

“Clear and unambiguous, professional English used throughout” -
Overall, the manuscript is generally clearly written although there are a few minor instances of grammatical/word-choice errors and a few passages that could be re-phrased to improve clarity and avoid repetition. The manuscript would benefit from being proof-read by a native English speaker, as there are minor language, grammatical and tense errors throughout.
Accessibility to readers of different expertise would be improved by defining the geometry mathematical terms used in their first instance - L37, “topology” (particularly as topology means something quite different in phylogenetic tree studies), L62, “polyhedral”, L96, “inscribed polygon”

“Literature references, sufficient field background/context provided.” -
I have a few suggestions for additional references that could be added to provide specific experimental or field results that support the authors more mathematical approach in addition to their own observations.
In general, I feel that the Introduction should provide some justification for this study, as none is provided. I have made some suggestions that the authors could include in comments under 2. Experimental Design.
I think that the discussion around the Results and Discussion section “Coccosphere contain only a single layer of coccoliths in this study” assumed the reader is well aware of the processes of cell division and its relationship with coccolithogenesis. However, not all readers will be and these processes are quite different from division cycles in other phytoplankton groups. For example, stating “The changing surface area due to photosynthesis, cell division and respiration” L75-76 is really too concise. To improve the clarity of the discussion, I think it would be worth the authors writing a few clear sentences that describe the cell division cycle, or to ‘pad out’ lines 74-79 with some additional linking sentences that would improve understanding. For example, the line I pulled out above could be precede by something like “Cell volume will increase during the course of the cell division cycle, initially cells are at their smallest immediately following cell division, then increase in diameter as organic carbon is fixed. During this time, each cell is sequentially producing coccoliths that join the coccosphere before cell division occurs again.” And then continue to say L74 “The changing surface area……”. To be honest, I think starting the Results and Discussion with this section is a little odd given that this is not the focus of your paper. If you adjusted the perspective, you could simplify your ideas into a few sentences and move it to the methods around L58-59 to say “Whilst E. huxleyi is well-documented for producing multi-layer coccospheres (references), we assume that the coccospheres in this study were comprised of just a single layer for the following reasons: 1…..2….3…..4….”

“Professional article structure, figs, tables. Raw data shared.”
The structure of the article is suitable and the figures are relevant and of sufficient quality for publication.
The authors have provided their raw data in the form of an excel file, which appears to include all of the raw data. However, the sheets should be more clearly labelled so that the reader know explicitly what every column contains rather than having to infer it from the equations in the sheet in many cases.

Experimental design

“Research question well defined, relevant & meaningful. It is stated how research fills an identified knowledge gap” -
The study states (L36-37) that the knowledge gap is “….the coccolith topology of coccosphere and the arrange mechanism remain unknown.” and their research question is stated as a result, that “(we) propose that cells arrange each of the coccoliths to interlock with 4-6 others to keep pace with cell growth and cell division”. Given this, I think that the authors should more explicitly state their research question, something along the lines of “to investigate the mathematical constraints that might underpin the arrangement of coccoliths around the cell surface in Emiliania huxleyi ”. Furthermore, the introduction would benefit greatly from a brief statement of why answering this question is of importance or general interest. I would suggest that understanding mathematical constraints on the arrangement of coccoliths around the cell surface would be interesting/important for (1) providing a theoretical context for empirical studies that have investigated the placement of coccoliths exuded to the cell surface, (2) understanding the maximum and minimum arrangements and their mathematical controls and limits would be extremely useful when trying to model the architecture of extinct species that we have only seen as loose coccoliths and never as intact coccospheres, which are rarely preserved in the fossil record, (3) some insight might be provided on the links between the rate of calcification, the duration of the cell division cycle and cell volume, (4) interesting insights using this approach may arise as to why certain species produce coccosphere of few, large coccoliths compared to coccospheres of many, small coccoliths. The authors might want to incorporate some or all of these aspects into their introduction, or other ideas they have, particularly as some aspects are introduced as discussion later on.

Throughout the Results and Discussion section, there is repeated reference to changes in cell size related to the cell division cycle. This is often poorly integrated or linked to previous or proceeding ideas and, as the cell division cycle is not really explicitly described for the reader in the first instance, can be confusing. Might I suggest that the Results and Discussion section might be more logical to following if there was a section specifically on ‘Size changes during the cell division cycle and its impact on the coccosphere’ that could neatly draw together the ideas in a coherent matter and replace the section “Changing surface area” that is difficult to see what the key messages are and doesn’t really fit.


“Methods described with sufficient detail and information to replicate” –
I have a few more specific comments relevant to the description of the methods:
The authors should add the growth rate that the cultures were growing at and state that their filters were taken during exponential phase of growth (as taking them during stationary would likely affect the amount of ‘malformed’ morphologies observed.
Were the filters taken at the same points in the cell division cycle, so as to minimize the spread of coccosphere geometries observed due to differences in what point of the division cycle cells were generally in, i.e., recently divided or due to divide imminently?
The authors should list in the methods section which ‘geometric parameters’ were measured and then refer to Figure 1. Also, was just a single coccolith measured on each coccosphere, or were multiple coccoliths measured? If so, how did the authors account for the curvature of the sphere/coccolith when measuring a 2D image if the coccolith was not lying flat on the upper surface?
L46 – I believe that “Aqual” should be “Aquil” as the medium name?
L46 – The Price et al. paper was published in 1989
L47-48 – could the authors please add the irradiance levels and light:dark cycle durations experienced under the two light conditions.
L63 – The authors should state what Euler’s polyhedral formula is here instead of or as well as later in the discussion, and a brief worded description that is not too technical would be advantageous to improve accessibility.
L64 – please also include the weblink URL where this software can be found, as requested by under “How to cite CeGe” on their webpage - https://www.math.uni-bielefeld.de/CaGe/
It is unclear to me how the measurements of the coccolith in cross-section illustrated in Figure 1b could have been measured from the SEM images, as you would have required an edge-orientated coccolith connected with the sphere but not interlocked with anything. Could the authors please clarify how these measurements were taken using the SEM, particularly the angles that would have been obscured by the curvature of the distal shields anyway?

Validity of the findings

No comment, all conditions are met.

Additional comments

These general comments are ordered by line, in addition to comments previously mentioned:

L34 – This sentence would benefit by expressing that E. huxleyi coccoliths vary in size between morphotypes, strains, within strain-specific populations, and even on individual cells.
L37 – Please define ‘topology’ in its first usage.
L37 – This sentence should read “…the coccolith topology of the coccosphere and the arrangement mechanism remains unknown”
L67 – this should be ‘Coccospheres contain….’
L68 – the addition of references to Hoffmann et al. (2015) and Sviben et al. (2016) would be appropriate here, as both studies use imaging technique that show ‘cut-throughs’ of the coccosphere, showing the layering.
L82 – this sentence should end “….found in the medium when culturing E. huxleyi ”
L67 -87 – in general, I felt that this whole discussion section was poorly structured and one idea did not flow smoothly or logically into another, with instances of repetition. The concepts are all there, but I think it would be expressed much more clearly with a degree of re-structuring of these idea and, as mentioned previously, additional sentences describing the cell division cycle. Or re-structuring and moving to the methods section.
L90 – As previously highlighted, I am unclear how you could measure everything shown in Figure 1b using the SEM. Please clarify in the Methods section.
L90-94 – can the authors provide a statement on how these values compare to other reported for E. huxleyi ?
L96 – please define ‘inscribed polygon’
L96-98 – you say “Due to the difference of coccolith size, it is obvious that the outer shields were only partly overlapped to interlock neighboring coccoliths”. However, many of the SEM images you show do have fully interlocked coccoliths and so this is not obvious to me. Also, I think that you would be more accurate to say “Due to the range of coccolith sizes…”. Can you report the range in coccoliths sizes observed on single coccospheres to give an idea of how much variability is observed?
L98-100 – I assume that this uses the means reported in Table 1? - did you test all of the coccolith length-width and central area combinations that were measured? Or can you calculate the actual proportions that would be required to result in an inscribed triangle existing in the area of the shield and then confirm that these measurement combinations were not observed? This might give an indication of whether inscribed triangles would exist in other mophotypes of E. huxleyi or only in other species, or only in circular-coccolithed species etc. It would seem to me that circular coccoliths with small tube cycles and wide shield widths would be able to have inscribed triangles.
L103 – could you express these as a percentage i.e., the 90th percentile is only 23% larger than the mean. It might better express that the variation is small.
L105 – add “need to be” between coccoliths would…..be much smaller”
L110 – Change to “Using these assumptions, the greatest possible number of edges…..”
L112 – replace ‘results’ with ‘SEM images’ and add theoretical between above and assumption and add an ‘s’ onto assumption.
L118 – finish this sentence with ‘…..or compromise its defensive abilities.”?
L121 – its interesting to note that the maximum number of coccoliths per cell that you observed was 30. Given a minimum of around 6, this would suggest that some cells have more than double the number of coccoliths that would be needed to fully cover two daughter cells following cell division (around 12). Do you have any ideas about this and whether these cells are still a single layer? I see from the data supplement that only 5 of 156 coccospheres had >20 coccoliths per cell so it might be worth mentioning this.
L141 – Interestingly, quite a few other extant and extinct species also have minimum numbers of coccoliths per cell at around 5-7 and coccospheres of 6 coccoliths forming cubic shapes have been reported in Helicosphaera carteri and extinct Toweius pertusus, an ancestor of E. huxleyi (http://www.mikrotax.org/Nannotax3/index.php?taxon=Toweius%20pertusus&module=Coccolithophores for images) and Umbilicosphaera bramletti – see the PhD Thesis of Sheward (2016) University of Southampton and the dataset of Gibbs et al. 2013 and Sheward et al. 2017.
L146-150 – I’m afraid that I don’t really understand what you’re trying to say in these lines, referring to removing coccoliths and then having them produced again one by one indicating that the coccosphere must have 4-6-edged faces. Please try to clarify this idea in a different way?
L160-161 – “Since the edge number of a coccolith must fall in the range of 4 to 6.” Is not a complete sentence.
L161 – remove the ‘, which’ and just write “Assuming a mother cell covered…” and continue on L162 “…..interlocked coccoliths that have divided into….”
L162-166 – Again, this discussion would be aided by a description of the cell division cycle earlier on.
L174 – also supported by data collected from modern and fossil coccolithophores in Gibbs et al. 2013, Sheward et al. 2017 and the thesis of Sheward (2016).
L171-204 – This whole section of the discussion, in its current form, doesn’t really fit well in the manuscript or make clear points. There is quite a bit of repetition of ideas and I’m really not sure what you are trying to tell the reader. It is confusing that you suddenly jump to discussing Braarudosphaera, which does has a fascinating coccosphere architecture, but is also completely different from E. huxleyi in the fact that it is an extracellular calcifyer and produces nannoliths not interlocking placoliths. The statement on L198 “….and thus pentaliths counld not interlock with others like the coccoliths or E. huxleyi ” is therefore completely redundant. I presume that the inclusion of Braarudosphaera in this section in “Changing surface area” is due to the difficulty of linking its coccosphere architecture with size changes relating to the cell division cycle?
L218 – ‘interlock’ should be ‘interlocking’
L224 – ‘arrange’ should be ‘arrangement’
L227 – should state somewhere that these ‘interlocking taxa’ are called ‘placolith’. Wuold be nice to add how many extant placolith species there are.
L227-228 – the genus names are mixed up here.
L230 – change ‘studied’ to ‘studies’
L235 – change ‘studies’ to ‘investigation’


Referenced additional studies:

Gibbs et al. (2013) Species-specific growth response of coccolithophores to Palaeocene–Eocene environmental change, Nature Geoscience
Sheward (2016) Cell size, coccosphere geometry and growth in modern and fossil coccolithophores, PhD Thesis, University of Southampton.
Sheward et al. (2017) Physiology regulates the relationship between coccosphere geometry and growth phase in coccolithophores, Biogeosciences
Sviben et al. (2016) A vacuole-like compartment concentrates a disordered calcium phase in a key coccolithophorid alga, Nature Communications.

Reviewer 3 ·

Basic reporting

The article explores an interesting subject – the geometry of coccolith arrangement. The English in the article could use work throughout; there are numerous grammatical errors, especially relating to the presence/absence of 'a' and 'the'. The citations are sufficient, the article is reasonably well structured, and the data is accessible, but there is very little explanation of what the reader is supposed to have learned that she could not have figured out from looking at some photographs of E. huxleyi, and very little explanation of what motivated the study and what interesting or useful results came from it.

Experimental design

It appears that the authors were careful in their methods, and the research falls within the aims and scope of the journal. However, it is very difficult to gather what is the research question the paper is addressing. Why are the results interesting? Why are the results not trivial? What was the hypothesis being tested?

Validity of the findings

The different software packages used in the manuscript should be described in much better detail so that the reader knows what is being done and how it relates to the question being investigated. Euler's polyhedral formula should also be described in much greater detail because it features prominently in the manuscript but is never properly introduced. I would not say that the study is described in enough detail to be reproducible. More importantly, though, I would say it's hard to determine what new information the authors were trying to show.

Additional comments

I think you can make this into an interesting paper but I think the writing needs a lot of work. To me it seems possible that you analysed many pictures of E. huxleyi and ended up with conclusions that had to be true and that are apparent just from looking at pictures of coccoliths. I think you need to do a lot more writing so that you can explain to the reader what exactly you did (so they could reproduce it if they wanted to) and why you did it, and what you learned about E. huxleyi that isn't already known.

---

## Round 0.2 · accepted · Accept

Thank you for revising the paper, which is now acceptable.

# Reviewer 3 ·

Basic reporting

no comment

Experimental design

no comment

Validity of the findings

no comment

Additional comments

The revised version is much clearer; I feel that all of my comments have been addressed satisfactorily.